# Blocking Myc to Treat Cancer: Reflecting on Two Decades of Omomyc

**DOI:** 10.3390/cells9040883

**Published:** 2020-04-04

**Authors:** Daniel Massó-Vallés, Laura Soucek

**Affiliations:** 1Peptomyc S.L., Edifici Cellex, 08035 Barcelona, Spain; dmasso@vhio.net; 2Vall d’Hebron Institute of Oncology (VHIO), Edifici Cellex, 08035 Barcelona, Spain; 3Institució Catalana de Recerca i Estudis Avançats (ICREA), 08010 Barcelona, Spain; 4Department of Biochemistry and Molecular Biology, Universitat Autònoma de Barcelona, 08193 Bellaterra, Spain

**Keywords:** omomyc, Myc, cancer, Myc inhibition, mouse models, peptides, anticancer drugs, new therapeutics

## Abstract

First designed and published in 1998 as a laboratory tool to study Myc perturbation, Omomyc has come a long way in the past 22 years. This dominant negative has contributed to our understanding of Myc biology when expressed, first, in normal and cancer cells, and later in genetically-engineered mice, and has shown remarkable anti-cancer properties in a wide range of tumor types. The recently described therapeutic effect of purified Omomyc mini-protein—following the surprising discovery of its cell-penetrating capacity—constitutes a paradigm shift. Now, much more than a proof of concept, the most characterized Myc inhibitor to date is advancing in its drug development pipeline, pushing Myc inhibition into the clinic.

## 1. Introduction

### 1.1. Myc

The Myc family of proteins (from now on Myc) is composed of three basic helix–loop–helix leucine zipper (bHLHLZ) transcription factors: MYC, MYCL and MYCN, also known as c-Myc, L-Myc, and N-Myc [1], which are functionally redundant in some contexts [2]. They belong to a larger network termed the Proximal Myc Network (PMN), composed of proteins with a bHLHLZ domain that allows dimerization and recognition of DNA [3].

Structurally, the Myc proteins consist of three differentiated regions: A transactivation domain (TAD) at the *N*-terminus, a central region, and the bHLHLZ at the *C*-terminus. Localized along its sequence, Myc contains four conserved regions known as Myc boxes (MB). The TAD controls transcription of Myc target genes. It contains MBI, which contributes to gene activation and protein degradation, and MBII, responsible for assembly of the transcriptional machinery that is critical for the majority of Myc’s functions. MBIII and MBIV are located in the central region. MBIII has been implicated in transcriptional repression, apoptosis, transformation, and lymphomagenesis. MBIV is implicated in transcriptional activation and repression linked to apoptosis and transformation, as well as modulation of DNA-binding. The bHLHLZ domain consists of the basic region, necessary for binding to specific DNA sequences called Enhancer boxes (E-boxes) (CACGTG) in the promoters of Myc target genes, and a helix-loop-helix leucine zipper domain, essential for the dimerization with its obligate partner from the PMN, the Myc-associated protein X (MAX) [4]. Myc contains two nuclear localization signals (NLS), one in the central region and one in the basic region. It does not homodimerize and is generally unstructured until heterodimerization with MAX occurs, causing the basic domains to adopt helical shapes that anchor to DNA major grooves (Figure 1A). Myc/MAX heterodimers recruit other cofactors that together stimulate RNA polymerases I, II, and III, switching on transcription of at least 15% of all genes, encoding both proteins and non-coding RNA products [5,6,7]. This global transcriptional regulation activates multiple processes including DNA replication, cell cycle progression, ribosome biogenesis, metabolism and mitochondrial biogenesis [5]. Myc is also in charge of the regulation of global chromatin structure, in part through upregulation of histone acetyltransferases [8,9]. Furthermore, it can also induce transcriptional inhibition through different mechanisms, the most frequently demonstrated being its interaction with Myc-interacting zinc finger protein-1 (MIZ-1) [10]. In this context, it is still unclear whether MAX is necessary for the formation of the Myc/MIZ-1 complex [11].

Other members of the PMN include MGA, MXD1 (MAD1), MXD3 (MAD3), MXD4 (MAD4), MXI1 (MXD2, MAD2), MNT, MLX, MLXIP (MONDOA), and MLXIPL (CHREBP). All these members dimerize with MAX, MLX, or both, and have different functions that can cooperate with Myc activity, antagonize it, alter gene expression directly and independently of Myc, or a combination of these mechanisms [3]. MAX, in addition to dimerizing with Myc paralogs, forms heterodimers with the MXD family (MXI1, MNT, and MGA), which can compete with Myc for binding to MAX and for E-box sites in shared target genes. Unlike Myc, MXD, MXI1, MNT, and MGA repress transcription through the recruitment of corepressor complexes and act as antagonists of Myc [15]. MLX forms dimers with MLXIP and MLXIPL, which can either support or antagonize MYC function depending on cell context [16].

In physiological conditions in adult tissues, Myc proteins are not present in quiescent cells but are rapidly induced in response to growth factor stimuli [1]. This switch is tightly regulated at the transcriptional and post-transcriptional level, and both Myc mRNA and protein display a very short half-life [17]. Also, Myc transcription is controlled by multiple extracellular and intracellular signals that funnel through an array of transcription factors, chromatin modifiers and regulatory RNAs that are either recruited to or synthesized at the Myc locus [18]. At the protein level, Myc can be activated, stabilized or degraded, and this occurs through several mechanisms including phosphorylation, ubiquitination, sumoylation, acetylation, and its association with different cofactors [19]. Therefore, in non-pathological conditions, Myc is ubiquitously expressed during embryogenesis, while in the adult it can be found in proliferative tissues or during regenerative processes like wound healing [20].

Importantly, besides promoting cell proliferation, Myc can also sensitize cells to apoptosis, adding an extra level of regulation to Myc-dependent processes and protecting the organism from unrestrained growth [21]. Myc induces apoptosis through different mechanisms. For instance, it has been reported that it indirectly activates the tumor suppressor ARF, which stabilizes P53 to induce apoptosis [22]. In fact, P53-induced apoptosis in response to DNA damage is dependent on endogenous Myc [23]. However, Myc also induces apoptosis through P53-independent mechanisms [24], one of them being MIZ-1-mediated transrepression. The interaction of Myc with MIZ-1 represses the anti-apoptotic *BCL-2* gene [25], required for MYC to induce apoptosis in some cellular contexts [26].

### 1.2. Myc in Cancer

Compared to other infamous oncogenes, Myc is rarely mutated in cancer [27], but, still, it is deregulated in most tumor types. Its aberrant expression is driven by several mechanisms at the DNA, RNA, and protein level. It is one of the most commonly amplified genes in human cancer [28], being the top copy number alteration in ovarian, breast and squamous cell lung cancer [27]. It is also often translocated to one of the immunoglobulin loci in multiple myeloma, Burkitt’s lymphoma and diffuse large cell lymphoma, or to T-cell receptor loci in T-cell acute lymphoblastic leukemia [29]. Finally, upstream oncogenic signaling from the Notch, Wnt, TGF-β, Hedgehog, EGFR, ALK, and Hippo pathways also drive aberrant expression of Myc in lung and many other cancers [17,30]. Therefore, even when Myc is not the driver oncogenic lesion, it acts as an integrator of extracellular and intracellular oncogenic signals, an attribute that makes it a ‘most wanted’ target for the treatment of cancer [31].

Indeed, Myc is estimated to be elevated or deregulated in up to 70% of human cancers [32]. Its oncogenic potential is unleashed when sustained aberrant Myc levels are coupled with loss of stress response checkpoints like Bcl-XL and P53, or induction by mitogenic signals like RAS. In these conditions, oncogenic Myc binds to strong canonical E-boxes at promoters, amplifying the output of existing gene expression programs, while also invading non-canonical, lower-affinity E-boxes at enhancers in a dose-dependent manner, resulting in ectopic regulation of previously silent genes [33,34]. Through this mechanism, Myc activates multiple gene programs involved in almost all the hallmarks of cancer: It is implicated in growth and proliferation [35,36], dedifferentiation and stemness [37,38], angiogenesis [39,40], migration and invasion [41,42], and even evasion of the immune system [43,44] and resistance to therapy [37,45].

### 1.3. Myc as a Therapeutic Target

Despite its role in cancer etiology and maintenance, though, Myc has not always been considered a tractable target for cancer therapy, due to both technical and conceptual concerns. More in detail:*Myc* and *Mycn* (but not *Mycl*) knockout mice are not viable and Myc has a role in several physiological processes linked to normal tissue regeneration [46,47,48], suggesting that inhibiting Myc could cause severe side effects for normal tissue homeostasis.The three Myc family members, MYC, MYCN, and MYCL, are partially-redundant transcription factors, so that Myc-inhibitory strategies, ideally, should target them all to obtain the most efficient therapeutic impact.Myc is an intrinsically disordered, non-enzymatic protein; hence, conventional small molecules that target highly conserved, fixed three-dimensional structures like ATP-binding pockets in kinases cannot be discovered or designed.Myc exerts its function in the nucleus of the cell, an elusive compartment to conventional therapeutics.Myc’s role in metastasis is controversial, linking it to both pro- and anti-metastatic activities [49,50] and suggesting that Myc inhibition could, in some contexts, favor metastasis development.

All these complications have hindered the development of Myc inhibitors for a long time and deemed Myc an “undruggable” target [51].

## 2. Omomyc Design and Characterization

### 2.1. Omomyc as a Myc Dominant Negative

When comparing Myc and MAX crystallographic structures, many similarities can be found in their dimerization domain, the bHLHLZ. However, MAX is able to both homodimerize and heterodimerize, while Myc can only form heterodimers with MAX. In 1998, we identified four charged amino acids located in the leucine zipper—the region of greater diversity between MYC and MAX—as an impediment to MYC homodimerization. They correspond to two glutamates (E61, E68) and two arginines (R74, R75), that display major steric and electrostatic clashes [52] (Figure 1B). That was the basis for the design of Omomyc, a 90 amino acid MYC mutant comprising MYC’s bHLHLZ with mutations in these four amino acids that alter its dimerization specificity. Glutamate 61 was substituted with a threonine (E61T) to provide better shape complementarity between the two monomers, while the other three amino acids were substituted with those present in MAX (E68I, R74Q and R75N) to remove repulsive charges at these positions [52] (Figure 1B). In a chimeric repressor assay using expression vectors coding for Omomyc, MYC, and MAX, we showed that Omomyc was able to homodimerize as efficiently as MAX, and to heterodimerize with both MYC and MAX [52]. These data were confirmed almost 20 years later by Jung et al., who showed that expression of Omomyc in U2OS osteosarcoma cells led to the presence of Omomyc homodimers and heterodimers with MYC and MAX in vitro. However, in contrast with previously published results, these authors showed that expression of exogenous MYC reduced the binding of Omomyc to MAX, but had little effect on homodimerization, pointing to a greater affinity of Omomyc for its homodimeric form [13]. Work by us and other groups also showed Omomyc to be an effective Myc dominant negative and included further co-immunoprecipitation (CoIP) experiments in human embryonic kidney 293T and small-cell lung cancer (SCLC) cells Lu135 and H2141, demonstrating the interaction of Omomyc with MYC, MYCL, and MYCN, therefore pointing to it as a potential pan-Myc inhibitor. Interaction with MAX and MIZ-1, but not the Myc antagonist MXD1 or other HLH proteins such as HEB, ID1, and HIF1-α was also detected [53,54]. This demonstrated the specificity of Omomyc for the Myc-MAX network, but not the rest of the PMN.

Our first electrophoretic mobility shift assays (EMSA) performed at physiological temperature demonstrated that Omomyc/MAX—and to a lower extent Omomyc/Omomyc—were able to bind E-boxes, and increasing concentrations of Omomyc reduced the DNA binding of MYC/MAX complexes [52]. We also suggested that Omomyc/Myc dimers would have low affinity for DNA [52]. EMSA results performed at 12 °C by Jung et al., in contrast, showed that Omomyc/Omomyc dimers bind DNA with higher affinity than the MYC/MAX ones, and that Omomyc/MYC dimers bind DNA with intermediate affinity, even higher than the one of MAX/MAX dimers [13]. Results on the ability of Omomyc/MAX dimers to bind DNA were inconclusive in that study.

Despite these differences, in all cases, Omomyc was shown to act as a dominant negative of Myc transcription function. Indeed, initial experiments in which Omomyc was expressed in the human kidney cell line BOSC-23 demonstrated that it was able to inhibit MYC-driven transcriptional activation and reduce the number and size of colonies of NIH/3T3 fibroblasts in a colony formation assay, interfering with Myc transforming function and decreasing cell proliferation [52]. Along the same lines, Jung et al. performed chromatin immunoprecipitation (ChIP) at the genome level in U2OS cells expressing Omomyc and physiological or exogenous MYC, and showed that overall binding patterns of MYC and Omomyc are very similar. In these assays, Omomyc appears to bind both consensus E-box (CACGTG) and non-consensus (CANNTG) binding sites [13]. ChIP sequencing (ChIP-seq) data revealed that Omomyc reduces binding of MYC at both promoters and non-promoter binding sites, and the decrease is more pronounced at E-boxes than in other regions. Surprisingly, Omomyc marginally reduced MYC binding to promoters that are already highly occupied by physiological levels of MYC and to which no additional MYC is recruited when MYC levels become elevated and “oncogenic”. This could be explained by the fact that MYC/MAX binding to these sites is presumably stabilized by the interaction with other transcriptional cofactors. In contrast, Omomyc efficiently decreases the recruitment of MYC to the promoters that are occupied upon supra-physiological levels of MYC. In terms of gene expression, quantitative PCR (qPCR) and RNA-sequencing experiments demonstrated that, in genes whose promoters are “invaded” by oncogenic MYC levels, Omomyc significantly attenuates both activation and repression by MYC. Finally, gene set enrichment analysis showed that Omomyc blocked the expression of Myc gene signatures, common to tumors characterized by high Myc expression [13].

Together, these results show that Omomyc is able to interfere with Myc-dependent transactivation through two different mechanisms: (1) Direct sequestration of Myc away from E-boxes and (2) competitive binding to E-box sequences as homodimer or as heterodimer with MAX. Since neither Omomyc nor MAX have a TAD, Myc target genes remain silent in the presence of the latter dimeric forms of Omomyc (Figure 1C).

### 2.2. Omomyc’s Role in Myc-induced Transactivation and Transrepression

In 2002, in a follow-up study to the first Omomyc design and in the attempt to better characterize Omomyc’s effect in cells, we made use of an Omomyc expression vector in C2C12 mouse myoblasts and Rat1 fibroblasts, with or without exogenous MYC expression. Unexpectedly, Omomyc enhanced MYC-induced apoptosis, while its expression in the absence of exogenous MYC did not cause any cell death [55]. Then, in 2004, we saw that overexpression of the anti-apoptotic gene *Bcl-x_L_* in Rat1 cells was able to impair MYC-induced apoptosis, but only in the absence of Omomyc [56]. Furthermore, this enhancement of MYC-induced apoptosis was found to be dependent on P53 and independent of ARF in mouse embryonic fibroblasts [56]. Since it was commonly believed that transrepression activity is responsible, at least in part, for Myc’s pro-apoptotic role [57], we surmised that the potentiation of Myc-induced apoptosis by Omomyc could indicate that Omomyc blocks transactivation, while instead enhancing transrepression of Myc target genes.

This hypothesis was verified by Savino et al. in 2011 [53], who shed some light into this dual behavior of Omomyc by performing luciferase reporter and ChIP assays on two Myc bona-fide target genes: *Nucleolin*, a Myc-transactivated gene through classical E-box binding [58], and *CDKN1A* (the gene encoding p21), a Myc-transrepressed gene through binding with MIZ-1 [59]. In 293T cells transfected with Omomyc and MYC expression vectors, Omomyc inhibited MYC-mediated activation of the *nucleolin* reporter in a dose-dependent manner without affecting its basal activity, and it did so by reducing the amount of MYC bound to the promoter, in part competing with it for direct binding to the E-box. In the case of the *CDKN1A* gene, though, Omomyc decreased the activation of the reporter in the same way as MYC did. In fact, when expressed together, Omomyc and exogenous MYC had a synergistic effect in the downregulation of *CDKN1A* activation. When analyzing the binding of MYC and Omomyc to the *CDKN1A* promoter by ChIP, Savino et al. showed that Omomyc increased the amount of MYC binding, and partially bound itself to the same promoter region [53]. Similarly, both *CDKN1A* mRNA [60] and p21 protein levels [54] were shown to increase upon Omomyc expression in colon carcinoma and SCLC cells, respectively.

It has been suggested that the explanation behind these edgetic properties of Omomyc may lie in its capacity to bind MIZ-1 [61]. In fact, Myc interacts with MIZ-1 through the HLH region [62], which is conserved in Omomyc, to repress gene expression. Therefore, Myc and Omomyc share this property and might induce apoptosis through a MIZ-1-dependent mechanism. Another explanation could be that Omomyc, by competing with Myc for binding to MAX, potentiates some of the MAX-independent functions of Myc [63], among them apoptosis [64]. In any case, Omomyc blocks most of the tumor-promoting functions of Myc but retains and potentiates some of the tumor-suppressive ones, making its mechanism of action a more potent anti-cancer strategy than complete Myc inhibition.

### 2.3. Omomyc and Epigenetic Markers

In order to evaluate whether Omomyc could also have an effect on Myc-dependent epigenetic modifications, Savino et al. transfected Rat-1 cells with vectors expressing MYC and/or Omomyc. In this experimental system, Omomyc clearly impacted histone 3 acetylation of lysine 9 (H3K9Ac) and methylation in the opposite way to MYC, acting again as a MYC antagonist, leading to decreased active and increased repressive chromatin marks [53]. In further work by some of the same authors, reduction in H3K9Ac by Omomyc was confirmed in U87MG glioblastoma cells [65]. In another study, Varnat et al. found both Omomyc and MYC to associate with protein arginine methyltransferase 5 (PMRT5), which correlates with glioma malignancy and poor survival, and induce histone 4 di-methylation of arginine 3 (H4R3me2s) [60]. The authors suggested that, despite both associating with PRMT5, MYC and Omomyc could exert a different functional effect. Indeed, inhibition of PRMT5 by a short hairpin RNA (shRNA) against its co-factor COPR5, restrained MYC transactivation, while recovering Omomyc-dependent repression of MYC targets [60].

### 2.4. Omomyc and Stemness

In a paper by Galardi et al. in 2016, the authors focused on the effects of Omomyc expression in cancer stem cells. Omomyc inhibited self-renewal, growth and migration of glioblastoma stemlike cells (GSCs) in vitro, by decreasing expression of genes involved in neural stem cell self-renewal and proliferation, including *SOX2*, *NOTCH1*, *CCNDI* (cyclin D1), and *NESTIN,* and increasing the expression of the tumor suppressor *PTEN*. In the presence of differentiation stimuli, Omomyc enhanced neuronal differentiation [66]. By ChIP-seq, it was found that Omomyc expression in GSCs and U87MG disrupted the binding of MYC at promoters, which were in turn occupied by Omomyc itself. Less than half of the U87MG peaks overlapped with those in GSC, consistent with the view that many Myc targets are cell-type specific. RNA-seq showed that Omomyc did not only attenuate a large number of mRNA transcripts, but also enhanced a similar number of them. Interestingly, Omomyc affected the microRNA (miRNA) expression profile too, repressing pro-tumorigenic miRNAs and increasing the expression of tumor suppressive ones. In particular, MYC-upregulated miRNAs (miR-17-92 and miR-106a/363 clusters) were decreased upon Omomyc expression, while MYC-downregulated miRNAs (miR-15a, -16, 23a, and -150) were increased. In this context, upregulation of miRNAs miR-200a/-429 by Omomyc caused repression of ZEB1, a protein associated with tumor invasion in glioblastoma [66]. Taken together, these results demonstrate that Omomyc not only interferes with proliferation, apoptosis and chromatin structure, but can also block other crucial Myc functions, such as self-renewal of cancer stem cells and invasion. Similar results of impairment of self-renewal ability upon Omomyc expression were also reported in glioma-derived mouse neuroprogenitor cells and patient-derived glioblastoma cells grown as neurospheres [67]. In addition, Varnat et al. showed that Omomyc downregulates *GLI1* expression in colon carcinoma cell lines, where *GLI1* encodes for a transcription factor responsible of inducing metastatic and stem-like phenotypes [60].

## 3. Omomyc as a Proof of Concept that Myc Inhibition is a Viable Therapeutic Option

### 3.1. Omomyc Efficacy In Vitro

The in vitro anti-proliferative and pro-apoptotic effect of Omomyc has been confirmed over the years in several cancer types, often by infecting multiple mouse and human cancer cell lines and patient-derived cells with doxycycline (dox)-inducible vectors that allow switchable expression of the dominant negative. These cancer cell lines include, for instance, pancreatic ductal adenocarcinoma (PDAC) [13], neuroblastoma [53], SCLC [54], glioblastoma [66,67], atypical teratoid rhabdoid tumors [68], non-small cell lung cancer (NSCLC), breast cancer and melanoma (unpublished data). In the case of SCLC, Omomyc suppressed growth of a panel of 9 cell lines harboring genetic inactivation of *TP53* and *RB1* and, in most of them, concomitant amplification of MYC, MYCL, or MYCN. Regardless of which member of the Myc family was amplified, their cell number was decreased upon Omomyc expression due to cell cycle arrest and/or apoptosis. Cell cycle arrest was induced either in G1 or G2/M and accompanied by activation of cyclin-dependent kinase inhibitors p21 and 27, and reduction of p16, all known modulators of G1/S transition. When these cell lines were treated with an shRNA against Myc, the observed Omomyc-dependent phenotype was faithfully recapitulated. Importantly, in this set of experiments, the cell line without detectable Myc levels showed the mildest response to Omomyc [54].

### 3.2. Omomyc Efficacy and Side Effects In Vivo

Given the indispensable role of Myc and Mycn during development, when we first genetically-engineered a mouse model to express Omomyc in 2004, we limited its expression to one tissue only. The chosen tissue was skin epidermis, where Myc had been shown to drive proliferation, but where concomitant Myc-induced apoptosis was innately suppressed [69]. Both Omomyc and a Myc-ER^TAM^ constructs were placed under the involucrin (inv) promoter, specific to keratinocytes. *inv-Myc-ER^TAM^* mice treated with 4-hydroxytamoxifen to activate Myc, quickly presented papillomatosis associated with hyperkeratosis and hypergranulosis, but double transgenic *inv-Myc-ER^TAM^/Omomyc* treated in the same way did not (Figure 2). Strikingly, skin in Omomyc-expressing mice preserved the normal epidermal keratinocyte differentiation program, even in the sustained presence of activated Myc, when it was accompanied by a high degree of apoptosis [56].

This study was the first demonstration that Myc inhibition by Omomyc has an anti-tumoral effect in vivo and did not seem detrimental to normal tissue homeostasis. However, two crucial questions still remained unanswered at that time: would Myc inhibition be effective in tumors where Myc was not the driver oncogene? And even if it did, would systemic Myc inhibition cause catastrophic side effects?

To answer these questions, we generated the first conditional ubiquitous Omomyc-expressing mouse: in this mouse, the Omomyc coding sequence was placed downstream of a tetracycline-responsive promoter element (TRE); the animals also harbored a reverse tetracycline transactivator (rtTA) under the promiscuous cytomegalovirus (CMV) or β-actin promoter. The resulting *TRE-Omomyc;CMV-rtTA* double transgenic mice expressed Omomyc in all tested tissues besides bone marrow upon administration of dox to their drinking water. These animals were then crossed with the well characterized *LSL-Kras^G12D^* mouse model of lung adenocarcinoma to test the therapeutic impact of a systemic Myc inhibitor. To our surprise, activation of the Omomyc transgene in tumor-bearing mice not only caused a decrease in proliferation, but also an increase in apoptosis and senescence, detected by Ki67, TUNEL, and β-galactosidase markers respectively, finally resulting in massive tumor regression [73].

Maybe even more strikingly and against preconceived notions, Omomyc expression did not cause any significant change in body weight, general animal activity, blood biochemistry or histopathological examination in low proliferative tissues (pancreas, kidney, liver, heart, and lung). Only skin, testis and intestinal crypts showed decreased proliferation. In skin, this resulted in a moderate thinning of the epidermis and inhibition of hair regrowth after shaving. In testis, Omomyc caused atrophy of spermatogonia and reduction in spermatocyte counts. The small intestine showed shortening of villi, although without increased apoptosis, and with complete maintenance of intestinal absorption and the barrier against bacterial infections. *TRE-Omomyc;β-actin-rtTA* mice showed the same phenotype [73,74] and were used to study the effects of Omomyc expression in bone marrow, where it caused a decrease in proliferation and onset of transient anemia and leucopenia. These symptoms were quickly compensated by extramedullary hematopoiesis in the spleen, except for very mild polycythemia. Importantly, withdrawal of dox and consequent abrogation of Omomyc expression restored normal cell proliferation of all tissues within one week and tissues became indistinguishable from untreated mice [73]. This manuscript, published in 2008, represented a real paradigm shift in the Myc field, being the first formal proof that Myc inhibition was feasible and extremely effective as a therapeutic approach against cancer, while being safe and well tolerated in normal tissues.

In a follow-up study in 2013, we focused on whether emergence of resistance to Myc inhibition by Omomyc could occur in mice bearing Kras^G12D^-driven lung tumors. We demonstrated that one cycle of 4 weeks of Omomyc expression was already sufficient to significantly extend survival in these animals. Importantly, tumors that re-appeared after doxycycline withdrawal were still sensitive to Myc inhibition, since addition of dox and reactivation of Omomyc eradicated newly-formed lesions. Based on these results, we proceeded to perform metronomic treatments with Omomyc (4 weeks on, 4 weeks off), and observed that, in these conditions, mice were kept alive indefinitely, showed no signs of severe side effects and presented a lower number of relapsing tumors after each round of treatment, until their complete eradication. When mice were euthanized after more than one year, the only tumors that were present in the lungs of dox-treated mice had silenced the Omomyc transgene, which suggests that the only way to evade Myc inhibition was to turn off its expression.

To address whether additional mutations might favor emergence of resistance, the same experiment was conducted by metronomic treatment of Kras^G12D^ tumors in a p53 defective background [71] with similarly dramatic results (Figure 2). Therefore, unlike for most therapies, no compensatory mechanisms can arise in these tumors to develop resistance to Myc inhibition, even in the absence of p53.

It is important to mention that RAS is able to extend Myc protein half-life by stabilizing it through phosphorylation [75]. Therefore, despite not being the driver oncogene, Myc plays an essential role in Kras^G12D^-driven tumors. In order to validate these findings in tumors arising in a different tissue and from a different oncogenic driver, and mostly to focus on whether Myc could have a role in tumor microenvironment maintenance, we made use of one of the best characterized mouse models of pancreatic islet tumors, developed by Doug Hanahan’s group. This model, called *RIP1-Tag2,* is driven by the simian virus 40 (SV40)-T antigen, and mice develop insulinomas as a consequence of T antigen expression under the control of the *insulin* promoter. These animals were crossed with two different Omomyc mouse models: *TRE-Omomyc;CMV-rtTA,* for systemic expression of Omomyc, and *TRE-Omomyc;RIP-rtTA*, for expression in the β-cell compartment only. Administration of dox to both models, with consequent activation of the Omomyc transgene, induced complete regression of insulinomas (Figure 2). This regression started first as collapse of the tumor microenvironment and involution of tumor vasculature. It occurred through exclusion of macrophages and neutrophils from the periphery and the interior of the tumor, respectively, abrogation of Vegf:Vegfr2 interaction, endothelial cell death, and hypoxia, and it was then shortly followed by apoptosis of β cells. Again, as previously observed in the lung model, at end point, the only tumors still present in dox-treated mice had silenced Omomyc transgene expression [70]. Hence, Myc in tumor cells was shown to be crucial for instruction of the tumor microenvironment and was confirmed to be a non-redundant node in cancer.

More recently and in the same line of reasoning, Omomyc expression in PDAC-bearing mice (*pdx1-Cre;LSL-KRas^G12D^;p53ER^TAM^;TRE-Omomyc;CMV-rtTA*) was shown to trigger quantitative regression of these highly aggressive tumors and of their extensive fibroinflammatory stroma (Figure 2), reiterating the point that Myc holds a key role in the coordination of the cross-talk between tumors and their microenvironment [72].

The idea that Myc could be a universal target in multiple tumor types was further elaborated in two more studies, in which Omomyc efficacy was assessed in different models of glioma: an Ha-Ras-driven, genetically engineered mouse model of invasive astrocytoma (*GFAP-^V12^Ha-Ras;TRE-Omomyc;CMV-rtTA*) and in patient-derived models of glioblastoma [66,67]. In the first case, we showed that, in *GFAP-^V12^Ha-Ras*-bearing mice, expression of Omomyc upon dox administration was able to both prevent the development of astrocytomas and reverse severe neurological symptoms associated with established disease, by decreasing proliferation, increasing apoptosis and causing mitotic defects, therefore reducing astrocytic cell density and increasing mouse survival [67] (Figure 2). In mice transplanted with patient-derived glioblastoma neurospheres harboring a dox-inducible Omomyc expression cassette, Omomyc expression recapitulated the same mitotic defects and greatly reduced intracranial cell density, conferring a survival advantage to the mice [66,67]. In addition, Galiardi et al. showed that Omomyc expression reduced the GSC marker OLIG2 in glioblastoma cells, as well as the number of glioblastoma migrating cells and vascularization of the tumor stroma. As previously reported for other models, a fraction of the cells in this model also silenced Omomyc expression and retained their tumorigenic features [66].

The study by Von Eyss et al. in 2015 added yet another model to the list of Omomyc sensitive tumors: the *MMTV-Wnt1* mouse model of breast cancer, where the authors demonstrated that Omomyc expression could induce a strong decrease in proliferation in breast cancer cells [76]. Additionally, in 2019, Alimova et al. showed that expression of Omomyc significantly reduces tumor growth and extends survival in an atypical teratoid rhabdoid mouse model [68], demonstrating for the first time its efficacy against pediatric cancer.

These experiments as a whole confirmed the non-autonomous effects of Omomyc on the tumor microenvironment, while also reinforcing the notion of its safety in long-term systemic treatments, and expanded the list of tumor types as candidate for Myc inhibition treatment, regardless of their tissue of origin or driving mutation.

## 4. From Proof of Concept to Pharmacological Approach

In the last two decades, the use of genetically-engineered cells and mouse models containing an inducible Omomyc construct has contributed extensively to our understanding of Myc biology. Furthermore, Omomyc revealed that Myc is essential for the maintenance and growth of multiple types of tumors, pointing to Myc as a universal target in cancer and proving that Myc inhibition is a safe and effective therapeutic strategy that is worth pursuing. However, until very recently, Omomyc was only considered a useful laboratory tool to study Myc perturbation but not translatable into a drug [61].

### 4.1. Recombinant Omomyc is a Cell-penetrating Peptide

The observation that other bHLHLZ proteins could behave as protein transduction domains [77], together with the fact that Omomyc contains an amphipathic helical basic region [52]—a common feature of cell-penetrating peptides [78]—encouraged us to test if the purified Omomyc mini-protein itself could be used as a therapeutic agent. To do so, Omomyc was recombinantly produced in *E. coli*, purified, characterized and tested for its cell-penetrating capacity and therapeutic efficacy against NSCLC models in vitro and in vivo [79]. Circular dichroism and nuclear magnetic resonance assays revealed that recombinant Omomyc forms homodimers and heterodimers with MYC and MAX, as for its transgenic counterpart. We hypothesize that, in cells, the equilibrium among the different dimers would also be determined by the relative abundance of the different monomeric species of the network. Interestingly, Omomyc homodimers were found to be more thermodynamically stable than MAX homodimers, and equivalent to Omomyc/MYC and Omomyc/MAX heterodimers [79], in contrast to some of the previously published data reported in Section 2.1 [13,52]. In the presence of DNA, thermal denaturation and fluorescence anisotropy experiments showed that both homodimeric Omomyc and heterodimeric Omomyc/MAX, but not heterodimeric Omomyc/MYC, bind canonical E-boxes at physiological temperature [79], confirming Omomyc’s originally described mechanism of action and in line with the EMSA results at 37 ºC shown in [52]. The basic region of Omomyc homodimers was also shown to assume the same contacts to DNA as MYC/MAX heterodimers [13], confirming the capacity of Omomyc to compete with MYC for binding to DNA.

By treating NSCLC, neuroblastoma, glioblastoma and melanoma cell lines with increasing concentrations of fluorescently-labelled recombinant Omomyc, we observed that Omomyc penetrated cells in a dose-dependent fashion, mainly through clathrin-mediated endocytosis and macropinocytosis (in addition to some caveolin-dependent mechanism), with a significant proportion of the protein localized in the nuclei [79]. Notably, macropinocytosis is a particularly interesting endocytosis mechanism for cancer treatment, since it is enhanced in cancer cells compared to normal cells and therefore could favor the targeting of Omomyc to tumor cells [80]. The mechanism of entry of Omomyc is highly dependent on its basic region, since an Omomyc mutant with reduced arginine content in this region is unable to penetrate cells [79]. This reinforces the notion that cell-penetrating peptides need a high content of positively-charged amino acids to efficiently cross the cell membrane, as previously reported [81].

### 4.2. The Omomyc Mini-Protein Behaves as its Transgenically-expressed Counterpart in Cancer Cells

Treatment of H1299, A549 and H1975 NSCLC cells with the Omomyc mini-protein in culture caused a reduction in total cell number in a dose-dependent manner, with 50% inhibitory concentrations (IC50s) in the low micromolar range, more effectively than in Myc-independent SH-EP neuroblastoma cells, recapitulating the effect of an siRNA against MYC. This proliferative arrest is reflected by changes in tumor cell cycle, as reported for transgenic Omomyc [79]. Microarray, ChiP-qPCR and Chip-seq analysis confirmed the on-target effect of the Omomyc mini-protein: it shuts down both transcriptional programs driven by Myc and gene signatures associated with poor prognosis in lung cancer, without affecting gene sets for other bHLHLZ transcription factors involved in the physiopathology of the disease. Omomyc causes this transcriptional reprogramming by displacing MYC from bona fide promoters, where it also impacts on H3K27 acetylation, and decreasing MYC occupancy throughout the genome, encompassing both MYC-specific, strong and weak motifs at superenhancers. In this study, at least partial displacement of MYC was observed in 97.3% of its active promoter regions [79].

Most of these data in cancer cells were confirmed in an independent study by Demma et al., who showed that recombinant and chemically synthesized Omomyc penetrated cells through an ATP-dependent mechanism, localized in their nuclei (in particular in the nucleoli) and showed in vitro efficacy in lymphoma and colon cancer cells with deregulated Myc. These cells responded with 50% inhibitory concentrations (IC50s) in the nanomolar or low micromolar range, while lymphoma cells with low Myc responded only at higher peptide concentrations [82]. By RNA-seq of colon carcinoma HCT116 cells treated with the Omomyc mini-protein, the authors confirmed the downregulation of Myc-driven gene signatures along with changes in the overall transcription profile. The downregulation of specific Myc targets such as *ASNS*, *SAT1*, *ID3,* and *EGR2*, as well as *CD274*—the gene encoding Programed death-ligand 1 (PD-L1)—was also verified by qPCR [82]. In the same study and in line with our own results, CoIP and proximity ligation assay showed that the Omomyc mini-protein forms homodimers and heterodimers with MYC and with MAX, and strongly reduces MYC/MAX dimers. Omomyc/MYC dimers are present in the cytoplasm, while Omomyc/MAX and Omomyc/Omomyc are present also in the nucleus, reinforcing the notion that Omomyc sequesters Myc away from the DNA and binds E-boxes together with MAX or as a homodimer [82]. By E-box DNA binding pulldown coupled with mass spectrometry in lysates from Omomyc-treated Ramos cells, the same authors also confirmed that Omomyc effectively competes with MYC and MAX for DNA binding, but also with other PMN members MXI1, MGA, and MXD3, and with Myc cofactors WDR5 and KMT2A. ChIP-qPCR was performed in Omomyc-treated HCT116 cells, showing once again that Omomyc displaces MYC from both high-affinity and low-affinity promoters, at least in part by binding to them. The authors suggest that Omomyc, unlike Myc, binds high and low affinity promoters with the same affinity, blunting the ability of Myc/MAX to bind and promote transcription from these sites. ReChIP assays in HCT116 cells showed that both Omomyc/MAX and Omomyc/Omomyc dimers were bound to chromatin [82].

In addition, two new observations were reported in this study: First, translating ribosome affinity purification and RNA immunoprecipitation assays showed that MYC, MAX, and Omomyc can interact with ribosomes and MAX RNA, indicating that their dimerization with MAX occurs cotranslationally. In the absence of Omomyc, MYC and MAX associate with translating ribosomes, but when Omomyc is present, the MYC ribosomal association is inhibited. Second, MYC is degraded through ubiquitination upon treatment with the Omomyc mini-protein [82].

### 4.3. Intranasal or Intravenous Administration of Omomyc is Safe and Efficacious In Vivo

Encouraged by all these in vitro results, we proceeded to in vivo studies. We first chose to directly deliver Omomyc mini-protein to the lungs of Kras-driven lung adenocarcinoma-bearing mice by intranasal administration. The mini-protein was already visible in the nuclei of some tumor cells after 4 h, and was still detectable in the lung tumors 24 and 48 h later. Importantly, 3 days of treatment with 2.37 mg/kg Omomyc reduced proliferation of tumor cells (detected by Ki67 positivity), caused a shutdown of several gene sets related to Myc activation and poor lung cancer prognosis, among others, and induced changes in chemokine and cytokine profiles, consistent with Myc’s role in modulating the tumor microenvironment. Four-week treatment with the same dose every second day completely stopped tumor growth and significantly reduced tumor grading when compared to vehicle-treated mice. This phenotype was the result, again, of reduced proliferation and induced apoptosis, but it was also accompanied by an influx of T lymphocytes, suggestive of immune reprogramming of the tumors [79].

Then, in order to unleash the full potential of Omomyc and test its applicability by systemic administration, we proceeded to intravenous administration. Here we would like to stress that, despite a report claiming that Omomyc declines rapidly in mouse plasma with kinetics that could limit its use in vivo [82], we observed that intravenous administration of Omomyc showed a half-life of Omomyc in the blood stream of 49 h and a plasma clearance compatible with the development of a drug with a reasonable dose regimen [79]. Critically, 4-week treatment with 60 mg/kg Omomyc 4 times a week of a NSCLC subcutaneous mouse model mutated in EGFR, PI3K and P53 (also resistant to erlotinib) caused a significant reduction in tumor growth, without any significant alterations in mouse weight, blood counts, biochemistry and light microscopy pathology reports of all major organs. In addition, in this particular model, combination with the chemotherapeutic agent paclitaxel proved superior to both stand-alone therapies and abrogated tumor growth, significantly extending mouse survival, without any added toxicity [79].

## 5. Ongoing Research and Future Directions

As expected, once Myc inhibition was shown to be both safe and extremely effective against different types of tumors, several laboratories around the world undertook the mission to develop their own anti-Myc therapeutic strategies. Among those, a notable example is Bromodomain and Extra-terminal domain inhibition (BETi), which has been used to indirectly inhibit Myc expression in some cellular contexts. Different approaches are instead based on reducing Myc translation or stability, while others use mechanisms of action more similar to Omomyc’s interference with Myc binding to E-boxes or blocking Myc’s heterodimerization with MAX [31]. Along this line, some groups have developed small molecules that bind to distinct sites in the bHLHLZ domain, and either prevent the formation of Myc/MAX heterodimers or distort the structure of pre-existing dimers so as to inhibit their DNA binding [83]. Recently, inhibitors of this class have shown promising in vivo efficacy [84].

Here, in the interest of this review, we would like to mention some companies and research laboratories that have used Omomyc itself to develop their own therapeutic tools (Table 1).

### 5.1. Omomyc Fusion with “Phylomers”

PYC Therapeutics (formerly Phylogica), linked Omomyc to a functional penetrating “Phylomer” peptide (FPPa) and tested the efficacy of this new entity against plasmacytoma, leukemia and breast cancer cells in vitro [86]. Similarly to what has been shown for Omomyc alone, FPPa-Omomyc blocked proliferation, induced apoptosis [85,86], disrupted MYC/MAX interaction, downregulated MYC-activated gene sets and de-repressed MYC-repressed ones. FPPa-Omomyc showed in vivo efficacy in a subcutaneous patient-derived mouse model of triple negative breast cancer (TNBC) when administered locally, decreasing proliferation, causing apoptosis, downregulating PD-L1, and extending mouse survival [85]. In this report, Omomyc alone did not induce those changes but was used at a much lower dose than in our studies in NSCLC models. Despite the encouraging results shown by FPPa-Omomyc, it should be noted that TNBC is a metastatic disease, so that local injection would clearly not be sufficient to reach all disease sites and other routes of administration would need to be explored. Also, lack of pharmacokinetics (PK) data and the high rate of apoptosis induced (close to 100% both in vitro and in vivo), could limit the pharmaceutical development of this approach due to low bioavailability or severe toxicity (Table 1).

### 5.2. Inclusion Bodies

A relatively unknown strategy to deliver anti-cancer proteins is in the form of bacterial inclusion bodies (IBs) [91]. IBs are amyloid-like insoluble aggregates of recombinant proteins that accumulate in the cytoplasm of bacteria as a by-product of their soluble, native-like counterparts, with which they co-exist. When cells are lysed and soluble recombinant proteins collected, IBs are usually discarded [92]. However, given the notion that they can attach and penetrate mammalian cells and act as slow release protein platforms, Pesarrodona et al. produced Omomyc-FNI/II/IV-H6 IBs in *E. coli* and tested their anti-tumor efficacy in TNBC models in vitro and in vivo [87]. FNI/II/IV binds CD44, a transmembrane glycoprotein implicated in cancer development and progression [93]. As expected, both Omomyc IBs and GFP control IBs were internalized into CD44+ TNBC cells in vitro through an endosomal route, but only Omomyc IBs had a cytotoxic effect on tumor cells [87]. In addition, the authors showed that, while weekly intratumoral injections of Omomyc IBs in a TNBC orthotopic cell-line derived mouse model caused no changes in tumor volume, they did increase p21 protein levels, suggesting some block of proliferation, and induced tumor necrosis [87]. As for FPPa-Omomyc, PK and systemic toxicity of Omomyc IBs should be evaluated before further development (Table 1).

### 5.3. Variants of Omomyc

In order to improve on Omomyc efficacy, several groups have worked on variants of the mini-protein.

For instance, Calo-Lapido et al. synthesized an 82-residue mini-protein featuring the DNA binding domain of MYC and Omomyc by Native Chemical Ligation. To avoid the formation of undesired products resulting from spontaneous deamination of asparagine residues N500 and N515, they were replaced by an A and a Q respectively, generating [AQ]MYC. A synthetic intermediate featuring cysteines at the ligation site (C523 C548) was also generated and named [AQ]MYC(SH) [88]. Equivalent versions of Omomyc, termed [AQ]Omomyc and [AQ]Omomyc(SH), were also synthesized and labelled with a fluorophore. Internalization experiments in A549 and HeLa cells showed that the cysteine-containing mini-proteins were 4 to 8 times better internalized [88]. However, to our knowledge, no further characterization of these peptides nor functional assays have been performed.

With the objective of specifically improving Omomyc’s affinity for E-boxes, Brown et al. developed a high-throughput *E. coli* expression workflow to generate and profile thousands of Omomyc analogs, followed by verification of a small number of the best mutants by solid-phase peptide synthesis [89]. Fluorescence polarization and direct fluorescence energy transfer assays were used to measure DNA affinity, and inhibition of cell proliferation of Ramos and/or HCT116 cells was used to measure efficacy. Out of thousands of candidates, a number of variants presented relatively higher affinity for DNA and/or some enhanced efficacy in vitro [89]. Additionally, these variants were reduced in size to obtain shorter versions of Omomyc lacking the coiled-coil domain and, according to the authors, still retained Omomyc’s anti-proliferative effect in HTC116 cells in vitro [89]. As for [AQ]Omomyc and [AQ]Omomyc(SH), no further characterization, PK, in vivo efficacy nor toxicity data for any of these variants has been reported yet (Table 1).

### 5.4. Other bHLHLZ Mini-Proteins

A recent publication has reported a recombinant 146 amino acid cell-penetrating mini-protein derived from the Myc antagonist MXD1 named Mad, which, even if not directly related to Omomyc, was compared to it and reported to be more potent [90]. Mad is composed of the mSin3a binding domain and the bHLHLZ domain of MXD1, with a S145A mutation to prevent phosphorylation and consequent ubiquitination. This variant retains MXD1′s capacity to bind MAX but not Myc, competing with the latter for MAX binding. Mad binds E-boxes in the promoters of Myc targets, blunting Myc binding to these sites and blocking Myc target gene expression. It also interacts with UBF, thereby repressing transcription of rRNA genes [90].

In this same study, Mad affected Myc target gene expression between 1.5 and 2 times more than Omomyc, despite having the same affinity for E-boxes [90]. However, when validating this reduction in selected genes by qPCR, the reduction in expression caused by Mad and Omomyc was the same. Surprisingly, treatment with Mad was 10 times more potent than Omomyc in reducing cell viability in the two cell lines tested (HCT116 and Ramos). This effect seemed to be dependent on Myc, since low Myc HDMYZ cells responded only to much higher concentrations of both Mad and Omomyc [90]. Validation in other cancer types and, importantly, efficacy, PK, and toxicity data in vivo will be key in understanding the translation potential of this protein to the clinic.

## 6. Conclusions

In the era of personalized medicine, a broad therapeutic that could be used against multiple types of cancer might seem a utopia. Nonetheless, Omomyc possesses several properties that point in the right direction: it induces apoptosis in cancer but not normal cells, blocks proliferation and invasion, is able to shut down the crosstalk between the tumor and its microenvironment, and triggers the recruitment of immune cells to the tumor site. However, most of these properties and the best therapeutic impact of Omomyc have been described when expressing it as a transgene in genetically-engineered cell lines and mouse models. Translating this proof of concept from the laboratory to the clinic is not an easy task, but our most recent findings show it is clearly feasible. The unexpected cell-penetrating properties of the recombinantly-produced, purified Omomyc mini-protein, its stability and capacity to reach tumor cells after intranasal and intravenous administration, together with its anti-tumor effects in preclinical models in vivo, have paved the way to clinical trials, which are expected to start in 2021, sponsored by the company Peptomyc SL. In the coming years we will learn about Omomyc’s PK and safety in patients and, most importantly, its efficacy, first against NSCLC and TNBC, and hopefully in many more cancer types to come. In the meantime, molecules derived from Omomyc or others that phenocopy its mechanism(s) of action are being—and will be—developed by us and others, increasing our chances of success in the ultimate goal: Developing the first clinically-approved Myc inhibitor, an urgent need for cancer patients. Notwithstanding, we believe that Omomyc should already be considered a success. It has taught us that Myc inhibition is a safe and effective therapeutic strategy that shares the advantages of targeted drugs—impacting more on cancer cells than their neighboring normal tissues—and at the same time overcomes their drawbacks—the emergence of resistance—by attacking a central non-redundant function in most, if not all, tumor cells. Whatever awaits from its clinical use will teach us more about Myc biology and the targeting of similar “undruggable” targets and, once again, we are looking forward to learning the lessons.

## Figures and Tables

**Figure 1 cells-09-00883-f001:**
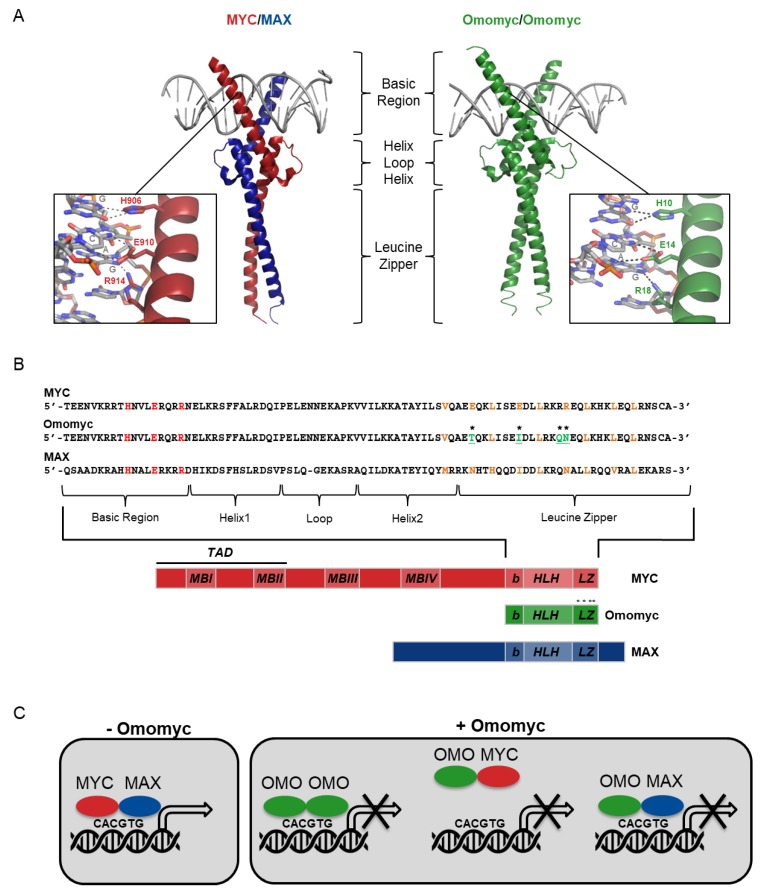
Omomyc acts as a dominant negative of Myc proteins. (**A**) Representation of the crystal structure of the MYC/Myc-associated protein X (MAX) dimer (1NKP, left) [12] and Omomyc/Omomyc dimer (5I50, right) [13] basic helix–loop–helix leucine zipper (bHLHLZ)-bound to DNA. Square boxes show a higher magnification of the basic region of MYC (left) and Omomyc (right) bound to a consensus E-box, with base-specific interactions as dotted black lines. PyMOL [14] was used to generate these representations. (**B**) Comparison between the sequences of MYC, Omomyc and the MAX leucine zipper. Residues that mediate the specific interaction of the basic region with DNA bases are colored in red, residues forming the hydrophobic core of the leucine zippers are colored in orange and the four mutated residues in Omomyc are colored in green. Below, a schematic representation of the MYC, Omomyc, and MAX proteins is shown. Asterisks represent the four mutated amino acids in Omomyc. (**C**) Representation of the interactions between MYC, MAX and Omomyc and their binding to DNA. When Omomyc is absent, MYC heterodimerizes with MAX and they together bind E-box sequences on the DNA, where MYC induces transcription of its target genes (left panel). When Omomyc (OMO) is present, it heterodimerizes with MYC sequestering it away from DNA, while also forming transcriptionally inactive homodimers and heterodimers with MAX that occupy E-boxes, resulting in inhibition of transcription of MYC targets (right panel). b: basic region. HLH: helix–loop–helix. LZ: leucine zipper. TAD: transactivation domain. MBI-IV: Myc boxes I-IV.

**Figure 2 cells-09-00883-f002:**
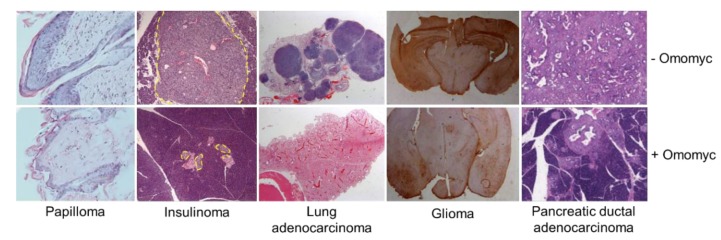
Expression of the Omomyc transgene has remarkable therapeutic impact in 5 different genetically engineered mouse models of cancer, from left to right: papilloma, *inv-Myc-ER^TAM^* (top panel) and *inv-Myc-ER^TAM^/Omomyc* (bottom panel), adapted from [56]; insulinoma, *RIP1-Tag2*;*TRE-Omomyc;CMV-rtTA* -dox (top) and +dox (bottom), adapted from [70]; lung adenocarcinoma, *LSL-Kras^G12D^*;*p53ER^TAM^;TRE-Omomyc;CMV-rtTA* -dox (top) and +dox (bottom), adapted from [71]; glioma, *GFAP-^V12^Ha-Ras;TRE-Omomyc;CMV-rtTA* -dox (top) and +dox (bottom) adapted from [67]; and pancreatic ductal adenocarcinoma, *pdx1-Cre;LSL-KRas^G12D^;p53ER^TAM^;TRE-Omomyc;CMV-rtTA* -dox (top) and +dox (bottom), adapted from [72]. All panels represent tissue sections stained with either an anti-GFAP antibody (glioma) or with hematoxylin and eosin (rest of the panels). Pancreatic islets in the insulinoma panels are circled with a dotted yellow line.

**Table 1 cells-09-00883-t001:** List of Omomyc variants and their features.

Type of Variant	Main Differences with Omomyc	Efficacy in vitro	Efficacy in vivo	Efficacy and Lack of Toxicity after Systemic Administration
**Omomyc [79]**		√	√	√
**FPPa-Omomyc [85,86]**	Efficacious at lower concentrations	√	√	Not reported
**Omomyc-FNI/II/IV-H6 inclusion bodies [87]**	Slow release Targeted to CD44+ cells	√	Some	Not reported
**[AQ]Omomyc(SH) [88]**	Enhanced cell penetration	Not reported	Not reported	Not reported
**Shorter Omomyc derivatives with enhanced DNA binding activity [89]**	Much shorter than Omomyc Not able to dimerize with Myc or MAX	√	Not reported	Not reported
**Mad (not derived from Omomyc) [90]**	Binds to MAX but not Myc Protected from ubiquitination More potent than Omomyc in vitro	√	Not reported	Not reported

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
