# Peer review of "Blocking Myc to Treat Cancer: Reflecting on Two Decades of Omomyc"

_cells, 2020, doi:10.3390/cells9040883_

Round 1
Reviewer 1 Report
The manuscript by Massó-Vallés and Soucek summarises the large body of work carried out in the last 20 years on Omomyc, a very interesting molecule which is long known to inhibit MYC hyper-activity through a dominant negative effect. I still remember the day I read the Soucek’s Oncogene paper first describing Omomyc, and it has been a real pleasure to read this well-written and well-structured review, which perfectly recapitulates its molecular and functional evolution up to become the most promising molecule restraining MYC activity in cancer. Moreover, the review introduces several aspects of MYC biology so that readers can easily understand why it is key in the maintenance of the whole cancer cell community in almost, if not all, cancer types, so as to represent the "perfect" target in cancer therapy.
Before publication in Cells, I recommend some minor additions/modifications, as follows:
- line 103: please add the Hippo pathway (with appropriate references) amidst the signalling cascades impacting MYC expression in human cancers.
- line 299, figure legend: I think there is something wrong in papilloma description, as genotypes are the same: is the element driving Omomyc expression lacking?
- lines 318-323: even if this cancer model is not directly driven by MYC dysregulation, it is well known that Ras mutations do extend MYC protein half-life by stabilising it through phosphorylation by dpERK. As most of the models the authors describe, though involving different organs, account on the expression of oncogenic Ras, they should mention this, just for accuracy.
- paragraph 4, lines 401-408: it was recently shown that MYC and JNK activities undergo reciprocal and inverse regulation in Drosophila and human cancer cells (Ma et al Oncogene 2017). Therefore, MYC deprivation, while reducing proliferation and growth, may promote migration. I think this is worth to be cited in this brief introduction as a “putative” side effect of MYC inhibition in cancer.
Author Response
Please, see the attachment.

Reviewer 2 Report
Overall, this is a very well-organized and comprehensive review of OmoMyc and its more recently developed analogs that nicely discuss the history of its discovery, the description of its anti-neoplastic properties and its mechanism(s) of action. Most of my criticisms are fairly minor:
- In Fig. 1, panel “A” should be so designated
- Lines 71-75. The explanation provided could be confusing for those who are not up to date in the Myc field. A diagram that shows the interaction between Max and either Myc or Mxds and between MLXIP and either (Mondoa A)/MXLIPL (ChREBP) and Mnt would be very helpful. Cross talk between the two pathways involving Mxd1, Mxd4 and Mnt should also be indicated.
- In order to provide context, the authors should include a section (perhaps best inserted after line 134) that briefly discusses the types of approaches that have been taken to inhibiting Myc with small molecules. At the very least, they should mention that these molecules bind to distinct sites in the bHLH-LZ domain, and either prevent the formation of a Myc-Max heterodimer or distort a pre-existing one so as to inhibit DNA binding (Fletcher S & Prochownik EV. Biochim Biophys Acta. 2015.1849:525–543). The recent description of more potent in vivo inhibitors of this class should be mentioned as well (Han H. et al. Cancer Cell. 2019.36:483–497). The concept behind OmoMyc is really not too different than its is for these molecules, i.e. a compound that prevents the formation of productive Myc-Max heterodimers and/or their ability to bind DNA.
- Line 181: The authors state: OmoMyc marginally reduced MYC binding to promoters that are already highly…”. It should be added that this is likely due to the fact that Myc-Max heterodimers are further stabilized upon binding to DNA and therefore that OmoMyc is likely able to disrupt this.
- There are some odd/awkward uses of English that warrant more careful editing by a native speaker. The following are just a few examples:
-Lines 174-175
-Lines 349-351
-Line 573
- The finding that the OmoMyc mini-peptide penetrates cells on its own is intriguing and seems to be the basis for its therapeutic effect. It is likely that the reason for this is because the basic domain of OmoMyc mimics some of the cell-penetrating peptides such as Tat, which tend to have a high content of basic amino acids. It would be a good idea to mention this and to speculate and whether adding an additional tag (perhaps on the C-terminus) might provide additional benefit. It may be that the discussed phylomer (line 517) represents an example of this but more information on this, how it compares with other cell-pentrating peptides and differences in efficacy seem warranted, particularly if speculating about the possibility of optimizing OmoMyc for in vivo delivery.
- The authors should also mention whether they have determined the kinetics of OmoMyc peptide uptake and its intracellular half-life.
- Do the authors have any idea why the OmoMyc peptide has such a long serum half-life? It’s small size would lead one to predict that it would be rapidly cleared by the kidney.
- If OmoMyc is as effective a penetrating cells as claimed by the authors, then it seems counterintuitive that it wouldn’t be taken up by nasal epithelial cells when attempting to deliver it in aerosolized form. Do the authors have any idea why the peptide is effective when delivered by this route? Similarly, when delivered by the IV route, why isn’t it taken up non-specifically by virtually all cells, especially by capillary endothelial cells where blood flow tend to slow considerably and or by the reticulo-endothelial system. Have any studies carefully assessed where in the body other than in tumors OmoMyc or its analogs is likely to concentrate?
Author Response
Please, see the attachment.
